# Low Body Mass Index Levels and Idiopathic Scoliosis in Korean Children: A Cross-Sectional Study

**DOI:** 10.3390/children8070570

**Published:** 2021-07-02

**Authors:** Kyoung-kyu Jeon, Dong-il Kim

**Affiliations:** 1Division of Sports Science, Incheon National University, Incheon 22012, Korea; jeonkay@inu.ac.kr; 2Sport Science Institute, Incheon National University, Incheon 22012, Korea; 3Health Promotion Center, Incheon National University, Incheon 22012, Korea; 4Functional Rehabilitation Biomechanics Laboratory, Incheon National University, Incheon 22012, Korea; 5Division of Health and Kinesiology, Incheon National University, Incheon 22012, Korea; 6Exercise Medicine & Disability & Rehabilitation Laboratory, Incheon National University, Incheon 22012, Korea

**Keywords:** cross-sectional study, Korean children, body mass index, idiopathic scoliosis, low body weight

## Abstract

Background: The prevalence of idiopathic scoliosis is rapidly increasing in Korean children, but research on the disorder is limited compared to that in other countries. Accordingly, in the present study, we aimed to investigate the relationship between idiopathic scoliosis and body mass index (BMI) levels in Korean children. Methods: This cross-sectional study enrolled elementary school students and middle school first graders in the Capital Area in Korea. The participants underwent body composition measurements and screening for idiopathic scoliosis. Idiopathic scoliosis was defined as a Cobb angle of ≥10°. The students were classified into three groups—the severely underweight (SUW: BMI < 16 kg/m^2^) group, the underweight group (UW: 16 ≤ BMI < 18.5 kg/m^2^), and the normal weight group (NW: 18.5 ≤ BMI < 25 kg/m^2^) to compare the risk of idiopathic scoliosis across BMI levels. Results: The final cohort comprised 1375 participants. The odds ratio (OR) of idiopathic scoliosis was 0.69 (95% confidence interval (CI): 0.50–0.94) and 0.66 (95% CI: 0.49–0.89) for the UW and the NW groups, respectively, with the SUW group as the reference. This shows that the risk decreased significantly by 31% and 34% in the UW and the NW groups, respectively. After controlling for age and sex, the corresponding ORs were 0.72 (95% CI: 0.52–0.98) and 0.70 (95% CI: 0.51–0.96), and the risk significantly decreased by 28% and 30% in the UW and the NW groups, respectively. Conclusions: Low body weight is closely associated with spinal deformity and idiopathic scoliosis.

## 1. Introduction

Scoliosis is defined as a Cobb angle of more than 10° on spinal radiography, a diagnostic criterion universally used since 1948 [1]. Idiopathic scoliosis causes physical deformities, including spinal distortion. This form of physical change can have negative psychological effects on children who are in a critical period of social development. Furthermore, it causes complex simultaneous problems related to musculoskeletal abnormality and social maladjustment [2,3]. As such, children should be actively screened for idiopathic scoliosis to prevent it. The prevalence of idiopathic scoliosis among Korean children increased by approximately 3.7 times between 2000 and 2008, from 1.66% to 6.17%, and continues to increase [4]. However, efforts to actively prevent and manage idiosyncratic scoliosis in children are limited in Korea.

More than 80% of scoliosis cases have unknown etiology, whereas the remaining 20% are caused by genetics and neuromuscular conditions [5]. The treatment and preventive strategies for scoliosis include exercise, bracing, and surgical intervention according to the patient’s Cobb angle, age, and health status [6], as well as healthy lifestyle habits and normal weight maintenance [7,8,9,10]. Studies conducted outside of Korea have reported that low body mass index (BMI) is closely associated with the occurrence of idiopathic scoliosis [7,8] and also negatively impacts physical and psychological health [9,11]. However, the impact of low weight and idiopathic scoliosis is currently overshadowed by the severe problems of reduced physical activity, overweight, and obesity in children in Korea. Maintaining an appropriate weight during childhood is critical in reducing the risk of scoliosis and maintaining healthy status, particularly because of growth spurts and the development of secondary sexual characteristics during puberty [9,10].

Although research on the epidemiology, the factors influencing psychosocial problems in children diagnosed with idiopathic scoliosis, and effective interventions to reduce spinal deformity is actively being conducted abroad [12,13,14], studies on children with idiopathic scoliosis are scarce in Korea. Accordingly, in the present study, we aimed to investigate the relationship between BMI levels and idiopathic scoliosis in Korean children ultimately to provide a basis for developing strategies to prevent and manage idiopathic scoliosis and promote lifestyle interventions for children with idiopathic scoliosis.

## 2. Methods

### 2.1. Study Subjects

This was a 2-year cross-sectional study that enrolled subjects randomly selected from elementary and middle schools in the Capital Area. We excluded those who had missing data on body composition analysis and scoliosis screening tests or who had a BMI over 25 kg/m^2^ (overweight). Some Year 1 data were analyzed in a study conducted to investigate the relationship between low weight and scoliosis in elementary school students [15]. In the present study, the relationship between obesity level and idiopathic scoliosis was more clearly identified with a sufficiently large sample by additionally recruiting study subjects in Year 2. All children who participated in the study were fully informed of the study objectives, and written consent was obtained from their parents. The study was approved by the Institutional Review Boards at Incheon National University IRB (No. 7007971-201612-003-01) (12 December 2016).

### 2.2. Measurements

#### 2.2.1. Body Measurements

Height, weight, and body composition were measured by trained researchers using a stadiometer (Seca, Germany) and a body composition analyzer (Inbody 720, Biospace, Korea) with the children wearing light clothing.

#### 2.2.2. Identifying Scoliosis

Due to the risk of radiation exposure during X-ray screening, many parents in Korea wish to avoid the use of X-ray screening for spinal deformities in their children. Therefore, we screened for the deformations of the spine using surface topography instruments in this study. Spinopelvic alignment was assessed through the use of spine structure analysis equipment (Formetric 4D, Dires International GmbH, Schlangenbad, Germany) using video rasterstereography. Specifically, 3D models were constructed based on the images obtained with triangulation by aiming a halogen light source on the surface of the child’s back (Figure 1). These models were then used in the assessment of spinal deformity. Several previous studies and meta-analysis studies have demonstrated the validity and reliability of the back surface measurement equipment [16,17]. Regarding the accuracy of the equipment, back surface images have a standard error of means under 0.15 mm, and spinal curvature measurements have a standard error of means within 3° [18,19].

### 2.3. Statistical Analysis

Statistical analysis was performed using SPSS/Window 25.0 (SPSS Inc., Chicago, USA). The independent *t*-test was used to compare differences in each measurement item between sexes. To investigate the effect of BMI on risk factors and the occurrence of idiopathic scoliosis, the participants were categorized into three groups according to their BMI levels as follows: the severely underweight group (SUW: BMI < 16), the underweight group (UW: 16 ≤ BMI < 18.5), and the normal weight group (NW: 18.5 ≤ BMI < 25).

First, an analysis of covariance with adjustments for age and sex was performed to compare the groups according to body composition and the condition of spinal deformity. Second, logistic regression analysis was performed with adjustments for age and sex to compare the odds ratios (ORs) of idiopathic scoliosis according to BMI. Statistical significance was set at *p* < 0.05.

## 3. Results

### 3.1. Patient Characteristics

In Year 1, 1062 children attending one of the three elementary schools participated in the study. In Year 2, 495 additional children were recruited from two elementary schools and two middle schools (13-year-olds, first graders only). Of the 1557 children in the study sample, 182 students (including 3 obese (25 ≤ BMI) students) who had not been administered tests evaluating body composition or scoliosis were excluded. Finally, the data from 1375 children were included in the analysis. The physical characteristics of the participants are shown in Table 1.

### 3.2. Effect of BMI on Spinal Deformity and the Risk Factors of Idiopathic Scoliosis

The different risk factors of idiopathic scoliosis according to BMI level in the 1375 Korean children are presented in Table 2. Compared to the UW and NW groups, the SUW group had significantly lower values of lean mass, body fat, sagittal imbalance, kyphotic angle, and lordotic angle. The scoliosis angle was significantly greater in the SUW group than that in the NW group (SUW: 12.05° ± 4.77°; NW: 10.96° ± 4.54°, *p* < 0.05); meanwhile, although the scoliosis angle was greater in the SUW group than that in the UW group, the difference was not statistically significant. Furthermore, boys showed significantly smaller angles of spinal curvature than girls (boys: 11.04° ± 4.49°; girls: 11.55° ± 4.81°, *p* = 0.041).

### 3.3. Risk for Idiopathic Scoliosis According to BMI

The risks of idiopathic scoliosis for each of the three BMI groups are presented in Table 3. The ORs of idiopathic scoliosis for the UW group and the NW group, with the SUW group as the reference, were 0.69 (95% confidence interval (CI): 0.50–0.94) and 0.66 (95% CI: 0.49–0.89), indicating that the risk of idiopathic scoliosis was significantly lower in the UW and the NW groups, by 31% and 34%, respectively. After adjusting for age and sex, the ORs were 0.72 (95% CI: 0.52–0.98) and 0.70 (95% CI: 0.51–0.96), showing that the risk was still significantly lower for the UW and the NW groups, by 28% and 30%, respectively.

## 4. Discussion

This cross-sectional study investigated the relationship of BMI with the risks of spinal deformity and idiopathic scoliosis in Korean children, where the incidence rate of idiopathic scoliosis among children is rapidly increasing [4]. The Scoliosis Research Society recommends that all children aged between 10 years and 14 years should undergo annual screening for scoliosis [20,21]. Early scoliosis screening in children is beneficial for the early detection and prevention of spinal deformation and is associated with a good prognosis [20,21]. Several developed countries, including the US, Japan, and some European countries, recognize the importance of scoliosis screening among children and conduct scoliosis screening at a national level, preventing and proactively managing spinal deformity and scoliosis in the child population [22,23,24]. However, active screening and preventive measures for idiopathic scoliosis in children are lacking.

Idiopathic scoliosis is closely associated with sex and obesity [8,25,26], and thus the present study examined the occurrence of scoliosis according to sex and obesity level. We found that compared to girls, boys showed significantly smaller angles of spinal curvature, which is the criterion for diagnosing scoliosis. Wong et al. (2005) observed the same trend regarding sex in their study of 72,699 children. They reported that scoliosis occurred in only 0.21% and 0.66% of boys aged 11–12 years and 13–14 years, respectively, whereas 1.37% and 2.22% of girls aged 11–12 years and 13–14 years had scoliosis. This shows that the incidence of scoliosis was approximately 6.5 and 3.3 times higher in girls aged 11–12 years and 13–14 years, respectively, than that in boys in the same age range [26].

Regarding the relationship between scoliosis and obesity level, the present study found that the risk of scoliosis was significantly lower in the UW and the NW groups than that in the SUW group in both adjusted and unadjusted analysis for age and sex. This finding is consistent with that reported by Yong et al. (2009) who studied 93,626 girls aged 9–13 years. In their study, the risk of scoliosis was higher by 1.5 times in the underweight group than that in the healthy and overweight groups [10]. Furthermore, in a study by Wang et al. (2016) that included 87 male adolescents, the BMI was significantly lower in the idiopathic scoliosis group (BMI: 18.5 ± 2.6 kg/m^2^) than that in the healthy control group (BMI: 21.3 ± 4.3 kg/m^2^; *p* < 0.001) [27]. Smith et al. (2008) studied 193 female adolescents (76 in the scoliosis group (BMI: 20.22 ± 2.92 kg/m^2^), 40 in the diabetes group (BMI: 23.27 ± 3.41 kg/m^2^), and 76 in the control group (BMI: 22.05 ± 3.39 kg/m^2^)) and found a significantly lower BMI in the scoliosis group than that in the other groups [8]. Recent meta-analysis studies [28,29] show that leptin levels in serum were significantly lower in adolescent idiopathic scoliosis (AIS) patients compared to healthy controls, and in terms of ghrelin levels, patients with AIS had higher levels of ghrelin than in healthy controls. Leptin and ghrelin have been reported to play important roles in regulating energy expenditure, body weight, and bone metabolism, which may affect the development of spinal deformities [30]. Collectively, these findings, including that of the present study, indicate that BMI is closely associated with idiopathic scoliosis. Specifically, the lower the BMI, the higher the risk for idiopathic scoliosis. It may also be proposed that the relationship between lower the BMI (underweight) and idiopathic scoliosis may be caused by hormonal problems such as imbalanced changes in leptin and ghrelin levels.

Lastly, lean mass was significantly higher in both the UW (14.60 ± 3.22 kg) and the NW (17.10 ± 3.90 kg) groups than that in the SUW (11.98 ± 2.10 kg) group. We speculate that being underweight and thus having a low BMI negatively affects muscle mass, bone density [31], and the stability and structural features of the musculoskeletal system, increasing the prevalence of spinal deformity and scoliosis.

The study has a few limitations. First, the study sample was limited to Korean children of a certain age range, and thus the findings cannot be generalized to other ages. Second, the study investigated correlation, rather than causality, of the risks of spinal deformity and idiopathic scoliosis and obesity level. Thus, the results should be interpreted cautiously. Lastly, the Cobb angle was investigated based on superficial topography and mathematical calculations using the spine structure analysis equipment (Formetric 4D) instead of X-rays in this study. Korean parents often avoid X-ray screening due to the risk of radiation exposure for their children. The spinal structural analysis equipment used in this study provides the ability to assess postures over time without the risk of radiation exposure. Therefore, adopting surface topography instruments has been proposed as alternative method of screening for spinal deformities instead of using X-rays. However, it provides a less accurate medical diagnosis than X-rays, and the results must be interpreted carefully. Despite these limitations, in this study we examined the risks of spinal deformity and scoliosis according to BMI level with a sufficiently large sample and particularly screened for scoliosis using highly reliable back surface measurement equipment, i.e., Formetric 4D [16,17,18,19]. Hence, this study is of significance in that it provides highly reliable results for the scoliosis status of the participants.

In conclusion, our results suggest that low body weight is significantly associated with the incidence of scoliosis. Particularly, our study emphasizes the association of low body weight with the prevalence of spinal deformity and idiopathic scoliosis. Further research should investigate the relationship between muscle mass and idiopathic scoliosis in Korean children.

## Figures and Tables

**Figure 1 children-08-00570-f001:**
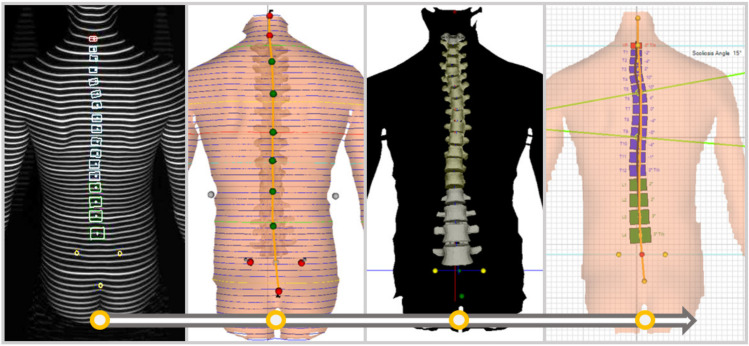
Spine structure analysis using video rasterstereography image of a patient’s back in the upright position.

**Table 1 children-08-00570-t001:** Subject characteristics.

Variables Total (*n* = 1375)	Male (*n* = 749)	Female (*n* = 626)	*p-*Value
Age (years)	10.86 ± 1.29	10.89 ± 1.28	0.686
Weight (kg)	39.38 ± 9.27	38.11 ± 16.13	0.069
Height (cm)	144.22 ± 10.08	143.72 ± 10.98	0.377
Lean Body Mass (kg)	15.84 ± 4.18	14.40 ± 3.34	<0.001
Body Fat (kg)	9.26 ± 4.74	9.69 ± 4.36	0.085
Percent Body Fat (%)	22.76 ± 8.36	24.88 ± 7.27	<0.001
BMI (kg/m^2^)	18.70 ± 2.70	17.93 ± 2.55	<0.001
Trunk Length (mm)	369.77 ± 34.04	373.46 ± 37.67	0.059
Sagittal Imbalance (°)	1.87 ± 2.97	1.70 ± 2.87	0.284
Coronal Imbalance (°)	−0.20 ± 1.52	−0.14 ± 1.49	0.418
Pelvic Inclination (°)	0.28 ± 2.98	0.24 ± 2.79	0.794
Pelvic Torsion (°)	0.78 ± 2.46	0.92 ± 2.26	0.309
Pelvis Rotation (°)	−1.50 ± 4.51	−1.23 ± 4.63	0.260
Kyphotic Angle (°)	42.39 ± 8.67	42.38 ± 9.35	0.994
Lordotic Angle (°)	34.55 ± 7.96	35.83 ± 8.05	0.003
Trunk Torsion (°)	−0.11 ± 5.87	0.34 ± 5.43	0.134
Scoliosis Angle (°)	11.04 ± 4.49	11.55 ± 4.81	0.041

Data and abbreviations are mean ± standard deviation. Abbreviations: BMI, body mass index.

**Table 2 children-08-00570-t002:** Variables according to BMI levels.

	Severely Underweight (BMI < 16)	Underweight (16 ≤ BMI < 18.5)	Normal Weight (18.5 ≤ BMI < 25)
Boys (*n* = 750)	*n* = 113	*n* = 272	*n* = 365
Age-Adjusted			
Scoliosis Angle (°)	12.41 ± 4.96	11.13 ± 4.24 *	10.55 ± 4.44 *
Girls (*n* = 625)	*n* = 157	*n* = 228	*n* = 240
Age-Sdjusted			
Scoliosis Angle (°)	11.79 ± 4.62	11.33 ± 5.14	11.60 ± 4.63
All Subjects (*n* = 1375)	*n* = 270	*n* = 500	*n* = 605
Age and Gender-Adjusted			
Scoliosis Angle (°)	12.04 ± 4.77	11.22 ± 4.67	10.97 ± 4.54 *
Lean Mass (kg)	11.98 ± 2.10	14.60 ± 3.22 *	17.10 ± 3.90 * ^
Body Fat (kg)	4.93 ± 1.51	7.31 ± 2.24 *	13.25 ± 3.89 * ^
Percent Body Fat (%)	17.24 ± 5.00	20.78 ± 6.00 *	29.07 ± 6.96 * ^
Trunk Length (mm)	357.25 ± 29.91	369.40 ± 37.12	379.46 ± 34.90 * ^
Sagittal Imbalance (°)	1.01 ± 2.92	1.70 ± 2.97 *	2.22 ± 2.82 * ^
Coronal Imbalance (°)	−0.23 ± 1.55	−0.21 ± 1.43	−0.12 ± 1.55
Pelvic Inclination (°)	0.28 ± 3.13	0.30 ± 2.71	0.22 ± 2.93
Pelvic Torsion (°)	1.20 ± 2.36	0.94 ± 2.25	0.60 ± 2.45 * ^
Pelvis Rotation (°)	−0.80 ± 5.01	−1.47 ± 4.55	−1.55 ± 4.36
Kyphotic Angle (°)	38.64 ± 9.47	41.44 ± 8.48 *	44.83 ± 8.45 * ^
Lordotic Angle (°)	33.02 ± 8.07	35.16 ± 8.39 *	36.02 ± 7.49 * ^
Trunk Torsion (°)	0.61 ± 6.98	0.02 ± 5.37	−0.09 ± 5.24

Data are mean ± standard deviation. Abbreviations: BMI, body mass index. * significant with 1st group, ^ significant with 2nd group, group *p* < 0.05.

**Table 3 children-08-00570-t003:** Odds ratios according to BMI levels.

		Severely Underweight (BMI < 16)	Underweight (16 ≤ BMI < 18.5)	Normal Weight (18.5 ≤ BMI < 25)
Boys (*n* = 750)		*n* = 113	*n* = 272	*n* = 365
OR (95% CI)				
	Scoliosis	1	0.73 (0.46–1.17)	0.50 (0.32–0.78)
Age-Adjusted OR (95% CI)				
	Scoliosis	1	0.76 (0.47–1.22)	0.53 (0.33–0.83)
Girls (*n* = 625)		*n* = 157	*n* = 228	*n* = 240
OR (95% CI)				
	Scoliosis	1	0.61 (0.40–0.94)	0.96 (0.63–1.47)
Age-Adjusted OR (95% CI)				
	Scoliosis	1	0.63 (0.41–0.97)	0.99 (0.64–1.54)
All Subjects (*n* = 1375)		*n* = 270	*n* = 500	*n* = 605
OR (95% CI)				
	Scoliosis	1	0.69 (0.50–0.94)	0.66 (0.49–0.89)
Gender-Adjusted OR (95% CI)				
	Scoliosis	1	0.69 (0.51–0.95)	0.67 (0.49–0.91)
Age and Gender-Adjusted OR (95% CI)				
	Scoliosis	1	0.72 (0.52–0.98)	0.70 (0.51–0.96)

Abbreviations: CI, confidence interval; OR, odds ratio; BMI, body mass index.

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
