# Peer review of "Low Body Mass Index Levels and Idiopathic Scoliosis in Korean Children: A Cross-Sectional Study"

_children, 2021, doi:10.3390/children8070570_

Round 1

Reviewer 1 Report

The authors present interesting and valuable research. The subject is important for the treatment of scoliosis. From the methodological point of view, there are problems, which should be further discussed.

  1. The authors in the introduction indicate that diagnostic criteria for scoliosis are Cobb angle > 10 on X-ray, but in the study, the Cobb angle was investigated based on superficial topography and mathematical calculations. Medical diagnosis and research must be done very precisely. Even when the correlation between DIERS and radiological measurements is very high, idiopathic scoliosis diagnosis is based on the radiography according to standards. The methodology should be as follows: 1. DIERS examination (screening) = suspicion of scoliosis 2. confirmation on X-ray and calculation the Cobb angle 3. assignment to the appropriate research group. The diagnosis of idiopathic scoliosis is not only the Cobb angle, it is also the exclusion of other causes of spinal deformity. The diagnostic process should be described in the research because the authors do not refer in the paper to the results of topographic measurements, but the clinical diagnosis of idiopathic scoliosis. Screening tests are definitely not enough to diagnose idiopathic scoliosis and relate it to the other examined parameters.
  2. It is not clear why the authors excluded obese children. If there is a significant correlation in the group of obese children, the conclusion that low body weight is related to scoliosis does not make sense. Unless every deviation from the norm is conducive to this risk.
  3. Tab 2. In the scoliosis angle line [there is no medical term of scoliosis angle, I understand that the authors mean the Cobb angle] in the SUW group, the Cobb angle is given: 12.41 +/- 4.96 which means, that the minimum angle is: 12.41-4.96 = 7.45 and this means no scoliosis (apart from the fact that it is a topographic measurement and mathematical analysis and not a radiological assessment). We do not know how many children in this group had a Cobb angle of less than 10 and should not be considered as children with scoliosis. The situation is similar in the group of girls and the UW and NW groups. Any measuring less than 10 degrees should be excluded from the group.
  1. Why is the risk calculated only for BMI and not for LBM, for example? It would be more interesting and easier to discuss the relationship between muscle mass and spine stability.

Author Response

I 'm full of gratitude for your review.
Please see the attachment.

Reviewer 2 Report

In present study, the authors investigated the relationship between idiopathic scoliosis and body mass index (BMI) levels in Korean children.  And they concluded that low body weight is closely associated with spinal deformity and idiopathic scoliosis.  As authors mentioned in the discussion section, the relationship between idiopathic scoliosis and body BMI levels has already reported in some journals, but it is new information that such tendency is also identified within the scoliosis patients in Korea.

The methodology of this study was precisely explained and acceptable.  Also, the limitations of this project were well indicated.

Therefore, I think this manuscript is appropriate for publication in Children journal.  However, I have a couple of minor requests to be considered as stated below.  After they have been resolved, I will judge this manuscript can be accepted and published by the Children journal.

*I ask the authors that correction parts will be shown in red color in the revised manuscript.

  1. Abstract line 23

I think “Low body weight are closely …” may be changed to “Low body weight is closely …”.

  1. Methods 2.1 study subjects line 61

A period is need after “Area” because here is the end of this sentence.

“… in Capital Area” should be changed to “… in Capital Area.”.

Author Response

(The authors gave the same response as above.)
